# SELF-HEALING: RECOVERING PRUNED LARGE REASONING MODELS VIA REINFORCEMENT LEARNING

## ABSTRACT

As large reasoning models (LRMs) achieve breakthroughs in reasoning tasks, building lightweight and efficient LRMs has become an urgent need for real-world applications. While structured pruning improves efficiency by reducing parameters, it often leads to significant performance degradation. To mitigate this loss, existing methods typically rely on next-token prediction, especially supervised fine-tuning (SFT) for recovery training. However, the effectiveness of pruning and recovery training in LRMs remains underexplored. Our empirical study shows that while structured pruning degrades the mathematical reasoning ability of LRMs, it does not completely destroy it, leaving room for compensation through recovery training. Existing recovery methods merely imitate reasoning trajectories in the training data, leading to performance bottlenecks and low data efficiency. To address this, we introduce reinforcement learning with verifiable reward (RLVR) for recovery training, enabling pruned LRMs to achieve self-healing performance. Experiments on five representative LRMs across six mathematical reasoning benchmarks show that RLVR significantly outperforms SFT-based recovery training. At 25% compression, RLVR-based recovery training improves performance from around 80% (with SFT) to over 95%, approaching or even outperforming the accuracy of unpruned LRMs while maintaining efficiency.

## 1 INTRODUCTION

In recent years, large language models (LLMs) have made significant breakthroughs in traditional NLP tasks (Bubeck et al., 2023). Research focus gradually shifts towards more challenging reasoning tasks. Nowadays, many large reasoning models (LRMs), including OpenAI-o1/o3 (OpenAI, 2025), DeepSeek-R1 (Guo et al., 2025), and Gemini-2.5 (DeepMind, 2025) have demonstrated extraordinary reasoning abilities. Reasoning, as a higher cognitive ability, often relies on emergent abilities enabled by models with tens or even hundreds of billions of parameters (Wei et al., 2022a). However, the high computational cost of such large-scale models severely limits their practical deployment. To promote the democratization of LRMs, researchers are exploring efficient reasoning models (Liu et al., 2025a; Li et al., 2025; Zhang et al., 2025; Wang et al., 2025b;a), to significantly reduce model size or computational overhead while maintaining reasoning performance.

Among various model compression techniques, pruning shows potential in reducing the computational demands of LLMs by removing model parameters (Frantar & Alistarh, 2023; Ma et al., 2023). In particular, structured pruning, which removes channels, blocks, or layers, can produce efficient models that do not rely on specific hardware. However, even the most advanced structured pruning methods still lead to significant performance degradation (An et al., 2024; Sandri et al., 2025; Gao et al., 2024), especially on complex tasks involving commonsense understanding and reasoning (Yang et al., 2024b). This limitation makes structured pruning still not widely applicable to lightweight LRMs. Therefore, post-pruning recovery training is essential to mitigate performance degradation and recover model capabilities. Designing an effective recovery mechanism is now crucial for practically deploying pruned LLMs or LRMs.

The mainstream recovery training relies on next-token prediction (NTP), including continued pre-training (CPT) and supervised fine-tuning (SFT). For example, Sheared LLaMA (Xia et al., 2024) uses large-scale corpora for continued pre-training, while LLM-Pruner (Ma et al., 2023) conducts SFT with instruction data to restore pruned model performance at a low training cost. Although these

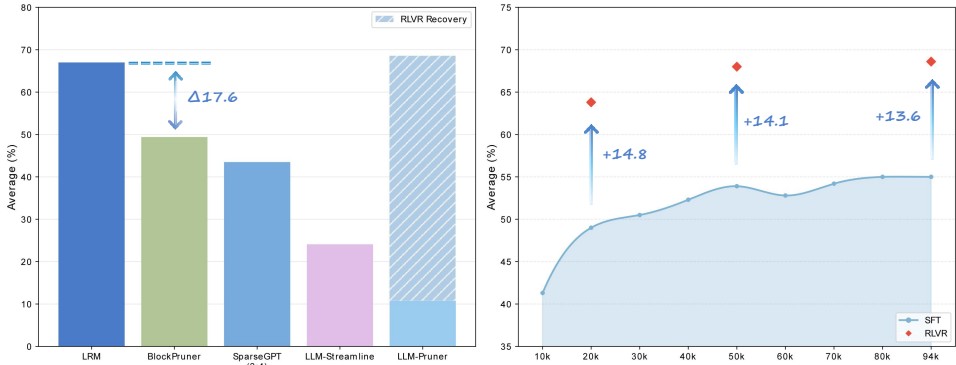

Figure 1: (Left) Structured or semi-structured pruning severely degrades the reasoning ability of DeepSeek-R1-Distill-Llama-8B, but the proposed RLVR recovery training can effectively recover it. (Right) RLVR can break the performance bottleneck in SFT recovery training for pruned LRMs.

methods can alleviate the impact of pruning to some extent, they still face critical challenges. First, there is a clear performance bottleneck. When the pruning ratio exceeds 20%, the model typically recovers only about 80% of its original performance, which limits the practicality of highly pruned models. Then, both CPT and SFT demand large-scale data, resulting in low sample efficiency. Moreover, existing pruning and recovery methods have not been thoroughly evaluated on LRMs. Therefore, whether they can be used to build effective lightweight LRMs remains an open question.

This work systematically investigates the impact of structured pruning on the mathematical reasoning ability of LRMs, and explores the potential of recovery training to mitigate performance degradation. Experimental results show that although pruning causes severe degradation in reasoning performance, it does not eliminate the mathematical reasoning ability, leaving room for recovery through appropriate training. However, mainstream SFT-based recovery suffers from both low data efficiency and a performance bottleneck. Moreover, by imitating the reasoning trajectories in training data, it substantially increases reasoning length, which ultimately reduces the efficiency of pruned LRMs. These limitations suggest that an effective recovery method should go beyond imitation, enabling the model to actively explore and strengthen its underlying reasoning capability. Therefore, we propose using reinforcement learning with verifiable reward (RLVR) to recover pruned LRMs. RLVR enables a self-healing process, allowing the pruned model to refine and explore its reasoning behavior. Considering the characteristics of recovery training, our approach balances exploration and exploitation in the design of rollouts and reference models. At the 25% compression rate, RLVR improves the average performance on six math reasoning benchmarks from 80% (with SFT) to more than 95% of the original performance, even surpassing the original LRM.

In summary, our contributions are as follows:

- We present the first comprehensive study on the impact of structured pruning on the mathematical reasoning ability of LRMs and the limitations of existing recovery training.
- This work is the first to introduce RLVR for recovery training, enabling pruned LRM to self-heal through exploration, thereby overcoming the limitations of existing recovery training methods.
- Experiments show that under a 25% pruning ratio, RLVR recovers over 95% of the math reasoning performance of pruned LRMs, while achieving better data and inference efficiency than SFT.

## 2  EFFECT OF MODEL PRUNING ON THE MATHEMATICAL REASONING ABILITY OF LRMS

### 2.1  DOES PRUNING COMPLETELY ELIMINATE THE REASONING ABILITY OF LRMS?

**Experimental settings**  We prune 25% of the parameters of DeepSeek-R1-Distill-LLaMA-8B using 3 structured pruning methods, including LLM-Pruner (Ma et al., 2023), BlockPruner (Zhong et al., 2025), and LLM-Streamline (Chen et al., 2025). We mainly focus on the mathematical reasoning ability of LRMs, and therefore utilize commonly used mathematical reasoning benchmarks for

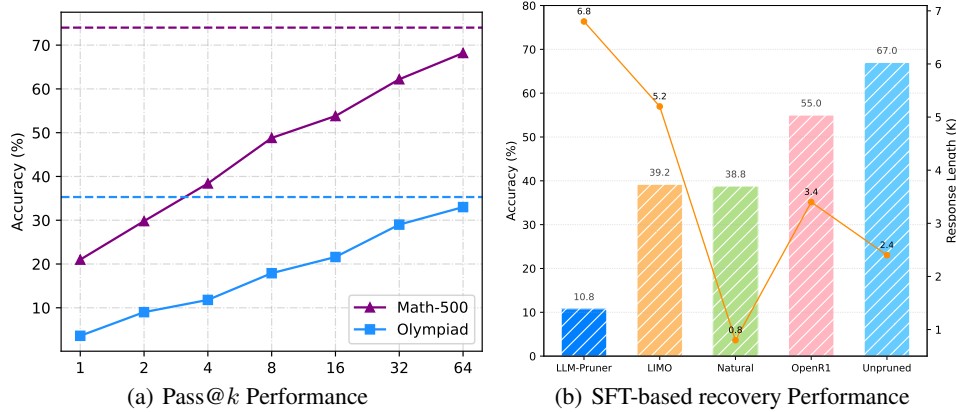

(a) Pass@$k$ Performance          (b) SFT-based recovery Performance

Figure 2: (a) Pass@k performance of 25% pruned DeepSeek-R1-Distill-Llama-8B on Math-500 and Olympiad Math. The dashed line indicates the performance of the unpruned LRM. (b) Performance and average response length of pruned LRMs after SFT recovery with different datasets.

evaluation, including GSM8K (Cobbe et al., 2021), MATH-500 (Hendrycks et al., 2021), Olympiad-Bench (He et al., 2024), College Math (Tang et al., 2024), Gaokao 2023 En (Zhang et al., 2023) and AMC 2023. During evaluation, we set the sampling parameters to the temperature of 0.6, top-$p$ of 0.95, top-$k$ of 20, and the maximum generation length of 8,192 tokens.

**Findings**  As Figure 1 (left) shows, pruning severely degrades the mathematical reasoning ability of LRMs. The state-of-the-art BlockPruner retains only 73% of the original performance, while LLM-Pruner preserve merely 16%. These results indicate that without recovery training, existing pruning techniques cannot serve as effective compression methods for LRMs. Then, we evaluate the reasoning ability retained in the pruned LRM. Specifically, we prompt the pruned LRM to solve each math problem $k$ times and calculate the proportion of problems for which at least a correct answer (i.e., pass@$k$). To encourage exploration of diverse reasoning paths, we set the sampling temperature to 1. As shown in Figure 2(a), although the pruned model performs poorly on pass@1, its pass@$k$ improves steadily with larger $k$, exhibiting an approximately log-linear growth. When $k$ is 64, its performance approaches that of the unpruned LRM. This indicates that **although structured pruning degrades the quality of individual outputs, it does not eliminate the LRM's underlying reasoning capability.** With a sufficiently large exploration space, the LRM remains able to generate correct reasoning results.

## 2.2 CAN SFT NEARLY LOSSLESSLY RECOVER THE REASONING ABILITY OF PRUNED LRMS?

**Experimental settings**  Following most structured pruning settings, we use SFT to recover the performance degradation caused by LLM-Pruner. We use three datasets from different sources for training: OpenR1[1], Natural Reasoning[2] (Yuan et al., 2025), and LIMO[3] (Ye et al., 2025). OpenR1 contains 94K high-quality mathematical reasoning problems along with reasoning trajectories generated by DeepSeek-R1. Natural Reasoning provides 1.1M multi-domain reasoning problems mined from pre-training corpora and synthetic trajectories. LIMO consists of 800 carefully curated and annotated mathematical reasoning examples, which have been shown to yield performance on par with reinforcement fine-tuning. LIMO is trained for 15 epochs and the other datasets for 1 epoch, with the learning rate of 5e-5 and the batch size of 32.

**Findings**  As shown in Figure 2(b), using OpenR1 as training data achieves the best recovery, achieving about 80% of the original performance. In contrast, the high-quality and small LIMO dataset recovers only about 60%, and the large-scaled Natural Reasoning performs similarly to

---

[1]https://hf-mirror.com/datasets/open-r1/OpenR1-Math-220k

[2]https://huggingface.co/datasets/facebook/natural_reasoning

[3]https://huggingface.co/datasets/GAIR/LIMO-v2

LIMO due to the low quality of its synthetic reasoning trajectories. For OpenR1, Figure 1 (right) further shows that as the training data increases, the recovery effect of SFT gradually saturates, leaving about 20% performance gap, which remains unacceptable in practice. Figure 2(b) also reports the average response length. Since SFT tends to imitate the reasoning trajectories of training data, models recovered with OpenR1 and LIMO produce longer responses than the unpruned model. In contrast, Natural Reasoning contains many short responses, and the recovered models fail to adaptively extend reasoning length according to problem difficulty, leading to poor performance. These findings indicates that **current SFT-based recovery methods are constrained by the quantity and quality of training data and suffer from performance bottlenecks, data and inference inefficiency** when recovering the mathematical reasoning ability of LRMs.

## 3 RECOVERY TRAINING VIA RLVR

To restore the mathematical reasoning capabilities of pruned language models, we adopt Group Relative Policy Optimization (GRPO) (Shao et al., 2024), a reinforcement learning method designed to enhance model performance while minimizing computational overhead. Unlike traditional Proximal Policy Optimization (PPO), GRPO eliminates the dependency on the critic model by leveraging group-normalized reward signals to estimate advantages. For a given question, GRPO samples candidate responses $\{o_1, o_2, \ldots, o_G\}$ from the previous policy model $\pi_{\theta_{old}}$, then computes token-level advantages using a reward normalization strategy based on group statistics. GRPO optimizes the policy model $\pi_\theta$ by maximization the objective below:

$$\mathcal{L}_{GRPO}(\theta) = \frac{1}{G} \sum_{i=1}^{G} \frac{1}{|o_i|} \sum_{t=1}^{|o_i|} \left( \min \left( r_{i,t} \hat{A}_i, \text{clip}\left(r_{i,t}; 1 - \varepsilon, 1 + \varepsilon\right) \hat{A}_i \right) \right) - \beta D_{KL} \left[ \pi_\theta || \pi_{ref} \right],$$

$$r_{i,t} = \frac{\pi_\theta(o_{i,t} \mid q, o_{i,<t})}{\pi_{\theta_{old}}(o_{i,t} \mid q, o_{i,<t})}, \hat{A}_i = \frac{r_i - \text{mean}(r)}{\text{std}(r)},$$

$$(1)$$

where $\pi_{ref}$ denotes the reference model, $\text{mean}(r)$ and $\text{std}(r)$ represents the average and standard deviation over rewards on $\{o_1, o_2, \ldots, o_G\}$, and $D_{KL}$ is a KL-divergence term. Most previous work sets the reference model as the initial policy and applies $D_{KL}$ to prevent the RL optimization from deviating too far. However, we argue that restricting policy updates in the recovery training setting may lead to suboptimal results. Therefore, we do not use a reference model and set $\beta$ to 0.

**Cold start via SFT** Since pruned LRMs suffer significant losses in instruction following and reasoning abilities, the zero RL training therefore suffers from severe reward sparsity, making the training ineffective. Therefore, we use the model trained by SFT recovery as the initial policy for RLVR recovery training. We choose math problem datasets with CoT as the training data. Given the training data $\mathcal{D} = \{q_i, r_i\}_{i=1}^{n}$, the minimization objective function is:

$$\mathcal{L}_{SFT} = -\frac{1}{\sum_{i=1}^{n} |r_i|} \sum_{i=1}^{n} \sum_{t=1}^{|r_i|} \log \pi_\theta(r_{i,t} \mid q_i, r_{i,<t}),$$

$$(2)$$

where $q_i$ is a question, $r_i$ is the response to $q_i$, including the CoT and answer. This objective encourages the model to imitate high-quality reasoning trajectories token by token.

**Temperature Warm-up for Exploration** A critical challenge in RLVR recovery is balancing stable exploitation and effective exploration. The training objective of SFT inherently induces a relatively fixed distribution. While a high decoding temperature can enlarge the exploration space, it fails to leverage the model's existing capabilities and may generate highly random reasoning trajectories, leading to sparser rewards and unstable gradients that hinder training stability. To overcome this trade-off, we adopt a temperature warm-up schedule. We start RLVR training with a low temperature, leveraging the stability and the reasoning behavior inherited from SFT. As training progresses, the temperature is gradually increased using a cosine warm-up function:

$$T(t) = \begin{cases} T_{\min} + A \sin\left(\frac{\pi}{2} \frac{t}{N_{\text{warm}}}\right), & 0 \leq t < N_{\text{warm}}, \\ T_{\max}, & t \geq N_{\text{warm}}, \end{cases}$$

$$(3)$$

where $T_{\min}$ and $T_{\max}$ are the minimum and maximum temperatures, $A = T_{\max} - T_{\min}$, $N_{\text{warm}}$ controls the warm-up steps. In this way, temperature warm-up allows RLVR recovery training to smoothly shift from exploiting the cold start policy to exploring better policy.

## 4 EXPERIMENTS

### 4.1 EXPERIMENTAL DETAILS

**LRMs** To evaluate the effectiveness of our proposed recovery training method, we conduct experiments on five LRMs: DeepSeek-R1-Distill-Llama-8B, Llama-3.1-Nemotron-Nano-8B, DeepSeek-R1-Distill-Qwen-7B, DeepSeek-R1-Distill-Qwen-14B, and Qwen3-8B. Among them, the DeepSeek-R1-Distill series are SFT-based reasoning models, whereas Nemotron-Nano and Qwen3 are trained with RLVR.

**Pruning** We use two structured pruning methods, width pruning LLM-Pruner (Ma et al., 2023) and depth pruning LayerDrop (Lu et al., 2024b), to perform experiments with 25% pruning rate. Among them, LLM-Pruner uses 16 samples from MATH with 2,048 tokens as calibration data and applies uniform layer pruning rates to facilitate recovery training. LayerDrop discards the last several layers without requiring calibration data.

**Training and Evaluation** We use the same settings as in Section 2.2 for the SFT cold start. In the RLVR stage, we utilize the verl (Sheng et al., 2024) framework to conduct GRPO on pruned LRMs. We adopt the rule-based reward function: the model receives a score of 1 for a correct answer and 0 for an incorrect one. We process prompts in batches of 1,024, generating 8 responses per prompt, with each response limited to 4K tokens. By default, the sampling temperature is 1.0, the clip ratio is set to 0.2, and the learning rate is 1e-6. The sampling temperature increased from 0.2 to 1.0 within 20 warm-up steps. We conduct all our experiments on 8 NVIDIA A100 GPUs. We use 8K math problems from MATH level 3-5 training data for 15 epochs reinforcement training. The evaluation settings are the same as described in Section 2.1.

### 4.2 MAIN RESULTS

**RLVR breaks the recovery performance bottleneck of SFT** We report the main results in 25% pruning ratio in Table 1 and Table 4. Overall, RLVR further improves the performance of recovery training on top of SFT, even recovering performance comparable to that of the unpruned LRM. For all five LRMs, width and depth pruning significantly degrade their performance on mathematical reasoning benchmarks. In particular, depth pruning reduces performance to less than 5% of the original LRM. Recovery training via SFT proves effective, except for Qwen3-8B, all other models can be recovered to about 80% of original performance through SFT with 94K long CoT data. However, the marginal benefit of collecting more high-quality SFT data for recovery training gradually decreases, making it difficult to compensate for the remaining about 20% pruning performance degradation by SFT alone. Our experiments show that RLVR may be the final step in narrowing the performance gap between pruned LRMs and their original models. For DeepSeek-R1-Distill-Llama-8B and Qwen3-8B, RLVR improves 15%-26% performance, fully recovering or even surpassing the unpruned models. For the other three models, RLVR achieves 94%–96% recovery. Although their mathematical reasoning ability is not fully restored, the remaining 4%–6% gap is acceptable given the trade-off between performance and efficiency. We believe this gap can be further reduced with longer RLVR training steps or sophisticated training tricks. Although RLVR is trained on only 8K simple math problems, its performance gains are not limited to basic benchmarks like GSM8K and Math. Similar improvements are observed across all evaluation sets, including more challenging benchmarks like Olympiad Math, College Math, and AMC 2023.

**RLVR is more data and inference efficient than SFT** It is widely acknowledged that the effectiveness of SFT depends heavily on the quality and quantity of training data. However, creating high-quality long CoT data is costly, relying on expert annotation or generation from advanced LRMs. Thus, improving either dimension is particularly challenging for long CoT data used in LRM training. In contrast, RLVR training only requires questions and ground-truth answers, without the need for detailed reasoning traces, making data acquisition significantly more straightforward and more

Table 1: Performance of various LRMs across multiple math reasoning in 25% pruning ratio. The best recovery performance method is indicated in **bold**. The gray percentages indicate the proportion of performance retained in the pruned model relative to the original model. Deepseek-LLaMA-8B and Nemotron-LLaMA-8B are abbreviations for DeepSeek-R1-Distill-Llama-8B and Llama-3.1-Nemotron-Nano-8B, respectively.

| Model | GSM8K | MATH-500 | Olympiad | College | Gaokao En | AMC23 | Avg. |
|---|---|---|---|---|---|---|---|
| Deepseek-LLaMA-8B | 85.6 | 80.6 | 46.1 | 36.2 | 73.5 | 80.0 | 67.0 |
| LLM-Pruner | 12.1 | 15.4 | 4.1 | 12.8 | 15.6 | 5.0 | 10.8 (16%) |
| + SFT | 78.5 | 72.6 | 36.0 | 32.6 | 65.2 | 45.0 | 55.0 (82%) |
| + RLVR | 86.0 | 83.3 | 51.5 | 41.4 | 74.2 | 75.0 | **68.6** (102%) |
| LayerDrop | 3.5 | 2.6 | 1.9 | 0.9 | 5.2 | 2.5 | 2.8 (4%) |
| + SFT | 81.5 | 76.8 | 38.8 | 34.7 | 67.8 | 50.0 | 58.7 (87%) |
| + RLVR | 91.0 | 83.4 | 50.8 | 39.7 | 76.6 | 75.0 | **69.4** (104%) |
| Nemotron-LLaMA-8B | 92.3 | 90.0 | 55.1 | 42.8 | 78.7 | 80.0 | 73.1 |
| LLM-Pruner | 7.4 | 10.6 | 1.6 | 9.7 | 10.6 | 7.5 | 7.9 (11%) |
| + SFT | 79.2 | 79.2 | 43.7 | 36.5 | 70.6 | 60.0 | 61.5 (84%) |
| + RLVR | 85.3 | 87.8 | 51.6 | 40.3 | 77.7 | 77.5 | **70.0** (96%) |
| LayerDrop | 4.7 | 5.2 | 2.6 | 1.8 | 3.5 | 2.5 | 3.4 (5%) |
| + SFT | 85.4 | 80.6 | 45.5 | 38.0 | 75.6 | 62.5 | 64.6 (88%) |
| + RLVR | 85.2 | 85.4 | 50.4 | 40.5 | 76.4 | 75.0 | **68.8** (94%) |

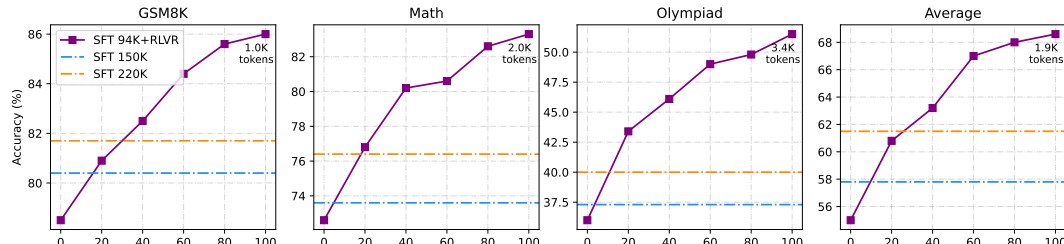

Figure 3: Recovery performance over the RLVR's training steps of the pruned DeepSeek-R1-Distill-Llama-8B. We show the performance on GSM8K, Math, Olympiad Math, and the average over all six benchmarks.

efficient. This gives RLVR strong potential for effective recovery in resource-constrained settings. Figure 3 compares RLVR after SFT with 94K data against SFT with extended OpenR1 datasets of 150K and 220K samples. The results show that while increasing SFT data beyond 94K yields further gains, the improvements are marginal. In contrast, RLVR surpasses the benefit of an additional 100K+ examples with only 8K samples and 20–40 training steps. Figure 3 also shows the average response length after RLVR training. Compared with the 3.4K tokens of SFT recovery in Figure 2(b), RLVR reduces the length by about 1.5K tokens, and by 0.5K tokens relative to the original LRM, demonstrating better inference efficiency.

## 5 DISCUSSION

### 5.1 HOW DOES TEMPERATURE WARM-UP SHAPE THE TRAINING DYNAMICS?

To evaluate the effectiveness of temperature warm-up and its underlying causes, we analyze the training dynamics of RLVR recovery with temperature warm-up and the fixed high temperature, where the latter follows Zeng et al. (2025) with temperature set to 1.0. Figure 4 illustrates that temperature warm-up in RLVR training not only reshapes the training dynamics but also achieves a better balance between performance and efficiency. In terms of overall accuracy, the warm-up strategy consistently outperforms the fixed-temperature baseline, both in the early and late stages of training. By the end of training, the fixed-temperature model plateaus at around 67% accuracy, whereas the warm-up strategy surpasses 69%. This suggests that dynamic temperature accelerates early performance gains and enables pruned models to achieve stronger recovery. For response

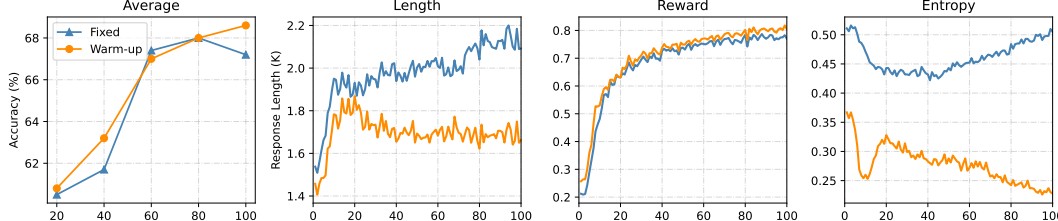

Figure 4: RLVR training dynamics on DeepSeek-R1-Distill-LLaMA-8B. Average denotes the model's mean performance across six math reasoning datasets, while Length, Reward, and Entropy refer to the policy model's average response length, reward, and rollout entropy during training.

length, the model trained with the fixed temperature tends to generate longer responses, from 1.4K tokens at the beginning to more than 2.1K tokens, while the warm-up strategy keeps the length stable around 1.7K tokens. This indicates that although fixed temperature encourages exploration, it also induces redundant reasoning trajectories and increases computational cost. In contrast, the warm-up strategy transitions from moderate exploration to concise and efficient reasoning trajectories. Reward and entropy curves further support this analysis. Temperature warm-up leverages reasoning ability inherited from SFT, producing denser reward signals and guiding better policy optimization. As a result, it sustains higher rewards even after both methods converge to the same temperature. Entropy trends reveal the exploration–exploitation balance. The fixed temperature sustains high entropy, reflecting persistent randomness without effective exploitation, while the warm-up strategy transitions from short-term exploration to effective exploitation of high-quality trajectories.

## 5.2 DOES RECOVERY TRAINING RELEARN OR AWAKEN MATH REASONING ABILITY?

As shown in Section 4.2, structured pruning severely degrades LRM reasoning, with some models retaining under 10% of their original performance. Yet, combining SFT and RLVR recovery restores over 95%. This raises an important question: does recovery training relearn or awaken mathematical reasoning ability? we compare the performance of pruned LRMs and LLMs under the same recovery training settings. As shown in Figure 5, after SFT recovery training, the LLaMA-3.1-8B-Instruct model, which lacks prior reasoning ability training, performs about 25% worse than the two LRMs. After RLVR recovery training, all three models show substantial improvement, but the performance gap between the LLM and LRMs does not narrow. This result further suggests that the effectiveness of recovery training depends not only on the training strategy itself

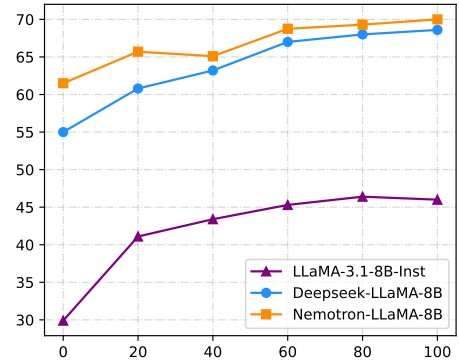

Figure 5: Comparison of the recovery performance of pruned LRM and LLM.

but also on the reasoning ability retained in the pruned model. Therefore, studying pruned LRMs is especially important. Compared to training a small model from scratch, an LRM obtained via pruning from a larger LRM is more likely to retain stronger reasoning capabilities.

## 5.3 HOW DOES RECOVERY TRAINING AFFECT THE BASIC CAPABILITIES OF PRUNED LRMS?

In LLM pruning research, most work evaluates the language modeling and commonsense knowledge capabilities. For LRMs, generalized abilities still need to be considered in practical applications. Therefore, we further investigate whether recovery training can also positively affect these foundational capabilities. Following standard evaluation settings in prior LLM pruning studies, we report perplexity on WikiText2 to evaluate language modeling ability, and average accuracy across seven commonsense understanding benchmarks (including BoolQ, Winogrande, PIQA, Hellaswag, ARC-e/c, and MMLU) to evaluate commonsense knowledge. As shown in Table 2, pruning significantly degrades both language modeling and commonsense performance.

Table 3: Performance of pruned DeepSeek-R1-Distill-Llama-8B in 50% pruning ratio.

| Model | GSM8K | MATH | Olympiad | College | Gaokao En | AMC23 | Avg. |
|---|---|---|---|---|---|---|---|
| Dense | 85.6 | 80.6 | 46.1 | 36.2 | 73.5 | 80.0 | 67.0 |
| *50% LLM-Pruner* | | | | | | | |
| SFT | 55.0 | 54.0 | 25.6 | 27.0 | 47.8 | 37.5 | 41.2 (61%) |
| RLVR | 81.1 | 69.3 | 36.2 | 34.7 | 61.6 | 52.5 | 55.9 (83%) |
| Two-stage | 81.0 | 69.1 | 35.7 | 34.8 | 62.0 | 50.0 | 55.4 (83%) |
| *50% LLM-Pruner + LayerDrop* | | | | | | | |
| SFT | 69.3 | 58.4 | 28.7 | 29.5 | 51.4 | 45.0 | 47.1 (70%) |
| RLVR | 86.4 | 72.6 | 41.1 | 34.5 | 66.3 | 50.0 | 58.5 (87%) |
| Two-stage | 86.2 | 72.6 | 41.5 | 35.9 | 65.7 | 65.0 | 61.2 (91%) |

Although our recovery training via SFT uses only math-related data, the results show that SFT can still partially recover language modeling and commonsense capabilities, especially in commonsense tasks, where performance improves by nearly 20%. In contrast, RLVR focuses on improving reasoning performance and has a slight impact on these basic abilities. In summary, if the practical application requires language modeling or commonsense understanding, we recommend supplementing recovery training with advanced SFT strategies (e.g., data selection or knowledge distillation). At the same time, RLVR can still be used as a powerful method to improve reasoning performance without harming these foundational capabilities.

Table 2: Performance of language modeling and commonsense understanding.

| Model | Wikitext | Commonsense |
|---|---|---|
| Deepseek-LLaMA-8B | 13.14 | 64.84 |
| LLM-Pruner | 3753 | 34.29 |
| +SFT | 22.29 | 47.80 |
| +RLVR | 22.27 | 49.44 |
| Nemotron-LLaMA-8B | 626.2 | 50.38 |
| LLM-Pruner | 1042 | 36.60 |
| +SFT | 29.75 | 45.82 |
| +RLVR | 29.54 | 47.39 |

## 5.4 Is recovery training effective for a higher pruning ratio?

In real-world applications, building lighter and more efficient LRMs requires applying higher pruning ratios. However, the aggressive pruning ratio always leads to more severe performance degradation. Can our proposed recovery training method remain effective under high pruning ratios?

We conduct experiments in the 50% pruning ratio with two different pruning configurations: 50% width pruning and pruning 50% across both width and depth dimensions. For the latter configuration, we prune parameters by 25% in width and 33% in depth. Table 3 shows the recovery performance in both settings. First, in terms of overall recovery, models pruned across both width and depth can achieve higher performance. This suggests that a more structurally balanced pruning strategy facilitates better recovery performance. Second, the recovery ability of SFT becomes significantly limited under high pruning ratios, recovering only 60%-70% of the original LRM's performance. In contrast, RLVR restores about 87%–91% of the original performance. Lastly, we compare one-shot pruning with the two-stage pruning strategy. While the two-stage method achieves slightly better performance, it also requires twice the training cost. Therefore, considering the trade-off between cost and gains, we do not recommend adopting this strategy in resource-constrained real-world scenarios.

## 6 Related Work

### 6.1 Large Reasoning Models

Large reasoning models, based on large language models, employ test-time or training-time strategies to enable human-like problem-solving in complex reasoning tasks. Chain-of-thought (CoT; Wei et al., 2022b) is the basic of LRMs, which prompts LLMs to generate intermediate reasoning steps before achieving the result. Empirical and theoretical works show that CoT expands the boundaries

of LLMs' reasoning capabilities (Feng et al., 2024; Li et al., 2024b). To break the error accumulation and linear thinking pattern limitation of vanilla CoT, researchers propose test-time compute strategies (Snell et al., 2025), such as repeated sampling (Wang et al., 2023; Cobbe et al., 2021; Li et al., 2023; Liu et al., 2025b), reflection (Shinn et al., 2023; Gou et al., 2024; Madaan et al., 2023; Kim et al., 2023) and tree search (Yao et al., 2023; Besta et al., 2024; Hao et al., 2023). On the other hand, training is also an effective way to evolve LLMs into LRMs. Supervised fine-tuning leverages human-annotated or LRM-generated long CoT data to improve weaker reasoning models (Ding et al., 2024; Tang et al., 2024). The strong performance of the Deepseek-R1-Distill series models highlights the importance of SFT. LIMO (Ye et al., 2025) and s1 (Muennighoff et al., 2025) assume that LLMs inherently possess reasoning abilities that can be activated with only around 1k training samples. Nevertheless, SFT still suffers from limited generalization and robustness (Li et al., 2024a), while reinforcement learning helps alleviate these challenges (Chu et al., 2025). For example, Qwen2.5-Math (Yang et al., 2024a) and DeepseekMath (Shao et al., 2024) use PPO and GRPO for post-training, respectively. DeepSeek-R1 (Guo et al., 2025) further demonstrates that even without SFT as a cold start, rule-based rewards alone can enable RL to unlock strong reasoning capabilities. Recently, many studies (Zeng et al., 2025; Dang & Ngo, 2025) have followed DeepSeek-R1's recipe to train LRMs on small and mid-sized open-source LLMs.

## 6.2 Recovery Training for Pruning Large Language Models

Pruning is a promising model compression method that reduces model size and accelerates computation by removing unimportant parameters. Based on the pruning granularity, it can be mainly categorized into unstructured pruning (Frantar & Alistarh, 2023; Sun et al., 2024; Yin et al., 2024; Zhang et al., 2024b; Dong et al., 2024; Ji et al., 2025), semi-structured pruning (Mishra et al., 2021; Zhang et al., 2024a), and structured pruning (Ma et al., 2023; Ashkboos et al., 2024; Ling et al., 2024). Unstructured and semi-structured pruning require specific GPU operator support for acceleration, while structured pruning, which does not depend on specific hardware, has attracted more attention. However, existing structured pruning methods for LLMs often result in severe performance degradation. As a result, retraining has become an essential step in most LLM pruning methods to compensate for the performance loss (Muralidharan et al., 2024; Xia et al., 2024). Chen et al. (2024) and Sengupta et al. (2025) conclude the scaling law for recovery training, showing that the final performance of the pruned LLM follows a power-law relationship with the number of tokens for recovery training. To achieve better recovery performance, Sheared LLaMA (Xia et al., 2024) uses the dynamic batch loading strategy, sampling recovery data at different proportions according to the loss on various domains. PASER (He et al., 2025) clusters data in the semantic space and adaptively selects appropriate proportions from each cluster based on the model's performance degradation, while filtering out conflicting or irrelevant recovery samples. Additionally, recent studies (Hu et al., 2025; Lu et al., 2024a; Munoz et al., 2024) have proposed mask-aware LoRA fine-tuning methods tailored to specific sparsity patterns in semi-structured pruning, reducing recovery costs while preserving the sparse structure. Existing recovery training methods mainly focus on improving LLMs' language modeling, commonsense understanding, and basic instruction following abilities, relying primarily on continued pretraining or SFT. Unlike previous works, our paper focuses on recovering more advanced reasoning capabilities in LRMs and explores the potential of RLVR in recovery training for the first time.

## 7 Conclusion

In this work, we systematically investigate the impact of structured pruning on large reasoning models and propose reinforcement learning with verifiable reward as an effective recovery strategy. Our study reveals that, although pruning substantially reduces model capacity, the mathematical reasoning ability of LRMs is not entirely eliminated, leaving significant potential for recovery. Compared with SFT-based recovery training, which primarily imitates training trajectories and suffers from performance bottlenecks and data inefficiency, RLVR leverages verifiable reward to guide recovery more effectively. Extensive experiments on five LRMs across six mathematical reasoning benchmarks demonstrate that RLVR consistently surpasses SFT, restoring pruned models to over 95% accuracy at the 25% pruning ratio and in some cases exceeding the performance of unpruned LRMs. These findings highlight the promise of RLVR as a self-healing mechanism for efficient LRMs, making it a promising direction for future research and optimization.

## REPRODUCIBILITY STATEMENT

To ensure reproducibility, we describe the detailed experimental settings in Sections 2 and 4.1. Furthermore, we provide the source code in the supplementary materials to facilitate independent verification of our results.

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

Table 4: Performance of various LRMs across multiple math reasoning in 25% pruning ratio. The best recovery performance method is indicated in **bold**. The gray percentages indicate the proportion of performance retained in the pruned model relative to the original model. Deepseek-Qwen-7B/14B is an abbreviation for DeepSeek-R1-Distill-Qwen-7B/14B.

| Model | GSM8K | MATH-500 | Olympiad | College | Gaokao En | AMC23 | Avg. |
|---|---|---|---|---|---|---|---|
| Deepseek-Qwen-7B | 93.2 | 88.1 | 51.1 | 41.3 | 78.5 | 78.3 | 71.8 |
| LayerDrop | 2.6 | 2.3 | 0.9 | 0.9 | 5.2 | 0.0 | 2.0 (3%) |
| + SFT | 83.4 | 78.0 | 38.7 | 37.7 | 67.8 | 50.0 | 59.3 (83%) |
| + RLVR | 85.7 | 85.4 | 48.3 | 42.1 | 76.4 | 75.0 | **68.8** (96%) |
| Deepseek-Qwen-14B | 92.7 | 89.0 | 52.9 | 40.3 | 80.0 | 87.5 | 73.7 |
| LayerDrop | 0.5 | 0.3 | 0.0 | 0.2 | 1.0 | 2.5 | 0.8 (1%) |
| + SFT | 85.1 | 70.6 | 36.7 | 35.6 | 66.5 | 65.0 | 59.9 (81%) |
| + RLVR | 92.0 | 84.2 | 47.9 | 38.6 | 72.2 | 80.0 | **69.3** (94%) |
| Qwen3-8B | 94.8 | 84.4 | 45.9 | 40.1 | 76.6 | 75.0 | 69.5 |
| LayerDrop | 5.5 | 3.0 | 3.6 | 1.8 | 5.2 | 5.0 | 4.0 (6%) |
| + SFT | 76.0 | 60.4 | 34.8 | 26.1 | 57.1 | 50.0 | 50.7 (74%) |
| + RLVR | 90.7 | 84.6 | 48.1 | 38.6 | 77.9 | 75.0 | **69.2** (100%) |

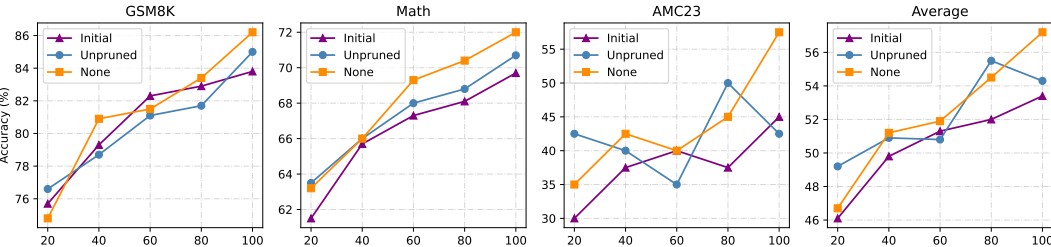

Figure 6: Recovery performance over the RLVR's training steps of the pruned DeepSeek-R1-Distill-Llama-8B. We show the performance on GSM8K, Math, AMC23, and the average over six benchmarks. "Initial", "Unpruned", and "None" represent using the initial policy, unpruned LRM as a reference model, and without a reference model, respectively.

Yuxin Zhang, Lirui Zhao, Mingbao Lin, Sun Yunyun, Yiwu Yao, Xingjia Han, Jared Tanner, Shiwei Liu, and Rongrong Ji. Dynamic sparse no training: Training-free fine-tuning for sparse LLMs. In *The Twelfth International Conference on Learning Representations*, 2024b. URL https://openreview.net/forum?id=1ndDmZdT4g.

Longguang Zhong, Fanqi Wan, Ruijun Chen, Xiaojun Quan, and Liangzhi Li. BlockPruner: Fine-grained pruning for large language models. In Wanxiang Che, Joyce Nabende, Ekaterina Shutova, and Mohammad Taher Pilehvar (eds.), *Findings of the Association for Computational Linguistics: ACL 2025*, pp. 5065–5080, Vienna, Austria, July 2025. Association for Computational Linguistics. ISBN 979-8-89176-256-5. doi: 10.18653/v1/2025.findings-acl.262. URL https://aclanthology.org/2025.findings-acl.262/.

## A    THE USE OF LARGE LANGUAGE MODELS

In preparing this paper, we used LLMs solely as a writing assistant to improve the fluency and readability of our manuscript. Specifically, LLMs were employed for grammar correction, phrasing refinement, and polishing of sentences. No parts of the research ideation, experimental design, or substantive content generation relied on LLMs. The authors take full responsibility for all scientific contributions presented in this paper.

## B    REFERENCE MODEL

The reference model is a key component in GRPO. In previous work, it is typically set as the initial policy model and used to constrain policy updates via a KL divergence penalty. This helps main-

tain training stability by encouraging exploration of high-reward regions while preventing the policy from drifting too far from its original distribution. However, as discussed in Section 3.1, this setting conflicts with recovery training. Recovery training's central goal is to compensate for performance degradation, which requires the policy model to move beyond the initial state and align more closely with the unpruned model. Thus, using the initial policy as the reference potentially hinders performance recovery. To address this contradiction, we explore two alternative reference models. The first uses the unpruned LRM as the reference model, serving as knowledge distillation, encouraging the current policy to align with a stronger model, and guiding recovery training in the right direction. The second approach removes the reference model entirely, eliminating the KL constraint and allowing the policy to fully explore and exploit its potential via RLVR.

Figure 6 illustrates the evolution of recovery performance under different reference model configurations with a 20k-long CoT cold-start during RLVR training. Overall, the choice of reference model significantly impacts recovery effectiveness. Using the initial policy model as the reference yields the weakest performance. In contrast, the unconstrained variant, trained without any reference model, achieves the best results, with an average performance gain of approximately 4%. Notably, employing the unpruned LRM as a reference model accelerates recovery in the early stages of training, helping the policy converge more quickly to a better solution. This guidance effect is particularly beneficial at the beginning of training, where alignment with the unpruned LRM helps the policy avoid poor local optima. However, as training progresses, this guidance gradually becomes a constraint. As the policy model continues to align with the unpruned LRM's distribution, its exploration space is increasingly restricted. Since the LRM's reasoning patterns are inherently complex and diverse, enforcing distributional similarity may limit the policy's ability to achieve further performance gains. In contrast, the unconstrained variant can continue to explore beyond the behavior of the reference model, ultimately achieving better performance. These findings suggest that reference models in recovery training should be used adaptively. Strong guidance is helpful during early recovery, while later stages benefit from relaxed constraints to foster exploration. This offers valuable insight for designing more flexible, dynamic reference mechanisms in future research.

## C CASE STUDY

We conduct the case study on the pruned DeepSeek-R1-Distill-Llama-8B after SFT and RLVR recovery training. The results show that both methods generate responses exhibiting fundamental reasoning patterns such as decomposition, verification, and reflection, with correct answers (marked in blue) appearing within the CoT. However, the model recovered with SFT shows a more chaotic reasoning process: although it initially generates the correct answer, it later rejects it during verification and reflection, ultimately producing an incorrect final answer (marked in red). In contrast, the model trained with RLVR demonstrates a clearer reasoning process. While there are still some redundancies, it retains the correct answer without overriding it in later steps and produces a final response that follows the verification format, despite no explicit format rewards during training. This highlights RLVR's advantage in improving both reasoning coherence and answer accuracy.

---

### Case 1

**Question:**
The set of points $(x, y, z)$ that satisfy 2x = 3y = -z is a line.
The set of points $(x, y, z)$ that satisfy 6x = -y = -4z is another line.
Find the angle between these lines, in degrees.

---

**Ground truth:** 90

---

**SFT:** Let's think step by step. <think> Okay, so I need to find the angle between two lines in 3D space. The lines are defined by the equations 2x = 3y = -z and 6x = -y = -4z. Hmm, let me recall how to approach this.

First, I remember that the angle between two lines is determined by the angle between their direction vectors. So, I need to find the direction vectors of each line and then use the dot product formula to find the angle between them.

Let me start with the first line: 2x = 3y = -z. This is a line in 3D space, and the direction vector can be found by looking at the coefficients of x, y, and z in the equations. Since all three expressions are equal, the direction vector should be proportional to the vector (2, 3, -1) ......Alternatively, ......But let me check ......Wait, ......But let me check if the direction vectors are correct. ......But let me check the calculation again. ......Alternatively, ...... But maybe there's a mistake here. Let me check with another approach. Let me parametrize both lines and find the angle between them.

First line: 2x = 3y = -z. Let me set 2x = 3y = -z = t. Then, x = t/2, y = t/3, z = -t. So, parametric equations: x = t/2, y = t/3, z = -t. So, direction vector is (1/2, 1/3, -1).

Second line: 6x = -y = -4z. Let me set 6x = -y = -4z = s. Then, x = s/6, y = -s, z = -s/4. So, parametric equations: x = s/6, y = -s, z = -s/4. Direction vector is (1/6, -1, -1/4).

Now, to find the angle between the two lines, we can take the direction vectors of each line and compute the angle between them. So, direction vectors are (1/2, 1/3, -1) and (1/6, -1, -1/4). Let's compute the dot product:

(1/2)(1/6) + (1/3)(-1) + (-1)(-1/4) = (1/12) - (1/3) + (1/4) = (1/12 - 4/12 + 3/12) = (0/12) = 0. Wait, that can't be. If the dot product is zero, the angle is 90 degrees. But that contradicts the previous result. What's going on here?

Wait, this is a problem. If the direction vectors are (1/2, 1/3, -1) and (1/6, -1, -1/4), then their dot product is zero. But that would mean the lines are perpendicular. But according to the previous calculation, the angle was arccos(1/(3 sqrt(14))) ≈ 85 degrees.

Predict: 85

---

**RLVR:**
Let's think step by step. Okay, so I need to find the angle between two lines given by these equations. Let me start by understanding what each line is. The problem says that the first line is defined by 2x = 3y = -z, and the second one is 6x = -y = -4z. Hmm, okay. So both are lines in 3D space, right? Because they have three variables x, y, z. So each equation here is like a parametric equation for the line.

First, maybe I should find the direction vectors of each line. Because once I have the direction vectors, I can use the dot product formula to find the angle between them. That makes sense. So direction vectors are key here. Let me recall that for a line defined by equations like ax = by = cz, the direction vector is (a, b, c). Wait, is that right? Let me check. ...... Alternatively ......

Alternatively, ...... Let me verify. ...... Wait...... Wait, ...... But this seems complicated, and the problem probably expects the answer to be 90 degrees. But why? Because the dot product is zero. So maybe the answer is 90 degrees. So perhaps the problem is designed so that the direction vectors are perpendicular. Therefore, the angle is 90 degrees. So maybe the answer is 90 degrees. Therefore, boxed answer is

boxed90. Wait, but let me check again. Maybe I made a mistake in direction vectors. Wait, if the direction vectors are perpendicular, then the angle is 90 degrees. So if the dot product is zero, then yes. So in that case, the answer is 90 degrees. Therefore, the angle is 90 degrees. So I think that's correct.

Therefore, the answer is $\boxed{90}$ degrees.

*Final Answer** $\boxed{90}$ </think>

......

Thus, the angle between the lines is $\boxed{90}$ degrees.

---

---

**Case 2**

**Question:**
What is the least positive integer multiple of 30 that can be written with only the digits 0 and 2?

---

**Ground truth:** 2220

---

**SFT:** Let's think step by step. <think> Okay, so I need to find the smallest positive multiple of 30 that only uses the digits 0 and 2. Hmm, let's start by recalling that a multiple of 30 must be divisible by both 3 and 10. That means the number has to end with a 0 (since it's divisible by 10) and the sum of its digits must be a multiple of 3 (divisible by 3).

First, since the number can only have 0s and 2s, the last digit has to be 0. So whatever number I come up with, it must end with a 0. Also, the digits can only be 0 or 2, so the number can't have any other digits. That simplifies things a bit because I don't have to worry about other digits interfering with divisibility.

Now, the number has to be the least such multiple. So I need to find the smallest number composed of 0s and 2s that ends with 0 and has a digit sum divisible by 3. Let's think about how to approach this. Since the number must end with 0, the last digit is fixed. The rest of the digits can be 0 or 2. The digit sum (sum of all digits) must be a multiple of 3. So, for example, if the number is 20, the digit sum is 2, which isn't divisible by 3. If it's 200, the digit sum is 2, still not divisible by 3. If it's 220, the digit sum is 4, not divisible by 3. 2020: sum is 4, same problem. 2220: sum is 6, which is divisible by 3. Wait, 2220 is a multiple of 30? Let me check.

2220 divided by 30 is 74, so yes, 2220 is 30 times 74. But is there a smaller number? Let's see. Let's check numbers with fewer digits first......

But wait, maybe there's a smaller number with more digits but smaller in value? For example, 2220 is 2220, but maybe a number like 2020 is smaller? Wait, 2020 is 2020, which is smaller than 2220. But 2020's digit sum is $2 + 0 + 2 + 0 = 4$, which isn't divisible by 3. So even though 2020 is smaller, it's not a multiple of 30. Similarly, 220 is 220, which is smaller than 2220, but digit sum 4. So 2220 is the first one that works. But let's check if there's a smaller number with more digits but smaller in value...... So maybe 2220 is the answer. But wait, let's check if there's a three-digit number. Wait, three digits: 200, 220, 2220?......

Predict: 220

---

**RLVR:**
Let's think step by step Okay, so I need to find the smallest positive multiple of 30 that only uses the digits 0 and 2. Hmm, let's start by recalling that a multiple of 30 must be divisible by both 3 and 10. Right? Because 30 is 3 times 10, so the number has to satisfy both divisibility rules. Let me confirm that. Yes, 30 is 2*3*5, so for a number to be divisible by 30, it must be divisible by 2, 3, and 5. Wait, but actually, since 10 is 2*5, and 3 is separate, so the number must be divisible by 10 (which requires a 0 at the end) and divisible by 3. So, the number must end with 0 (since it's divisible by 10) and the sum of its digits must be a multiple of 3. Got it.

So, the problem reduces to finding the smallest number composed of 0s and 2s that ends with 0 and has a digit sum divisible by 3. Let's think about how to approach this. ......

Wait, ...... Wait, but let's check.

Wait, 2220 divided by 30 is 74, so yes, it's a multiple. But is there a smaller number? Let's see. Let's think of numbers with two digits. ...... So, 2220 is the first one. So, maybe 2220 is the answer. Wait, but let's check if there's a smaller number with more digits. For example, 2220 is four digits, but maybe 2220 is the minimal. Wait, but maybe 2220 is the answer. Wait, but let's check 2220. Wait, 2220 is 2220, which is 2220. So, yes, that's the answer. Wait, but maybe there's a smaller number. Wait, let's think of numbers with digits 0 and 2, ending with 0, and sum divisible by 3. Let's try to find the minimal number. So, starting from the smallest possible numbers. Let's see. The minimal number would have the fewest digits. So, starting with two digits: 20. Sum is 2. Not divisible by 3. Three digits: 200. Sum is 2. Not divisible by 3. Four digits: 2220. Sum is 6. Divisible by 3. So, 2220 is the first one. Therefore, 2220 is the answer. So, the least positive integer multiple of 30 that can be written with only the digits 0 and 2 is 2220. Therefore, the answer is $\boxed{2220}$.

Wait, but let me check again. ...... Therefore, 2220 is the answer. So, I think that's the answer.

*Final Answer** $\boxed{2220}$ </think>

......

Thus, the least positive integer multiple of 30 that can be written with only the digits 0 and 2 is $\boxed{2220}$.

**Case 3**

**Question:**
In how many ways can 8 people sit around a round table if 3 of the people – Pierre, Rosa, and Thomas – all want to sit together? (Two seatings are considered the same if one is a rotation of the other.)

**Ground truth:** 720

**SFT:** Let's think step by step. <think> Okay, so I need to figure out how many ways 8 people can sit around a round table, with the condition that Pierre, Rosa, and Thomas all want to sit together. And rotations of the seating are considered the same. Hmm, let's start by recalling some basics about circular permutations.

First, for a circular table with n people, the number of distinct seating arrangements is (n-1)! because rotations are considered the same. So for 8 people, it would be 7! = 5040 ways. But here, we have a restriction: Pierre, Rosa, and Thomas must all sit together. So I need to adjust for that.

When there are groups that must sit together, we can treat them as a single unit or ̈block.S̈o instead of 8 individual people, we have 3 people in one block and the remaining 5 people. That makes a total of 6 units to arrange around the table. But wait, since it's a circular table, the number of arrangements for these 6 units would be (6-1)! = 5! = 120. But hold on, the block of Pierre, Rosa, and Thomas can themselves be arranged in different ways within the block. Since there are 3 people in the block, the number of ways to arrange them is 3! = 6. So we need to multiply the number of arrangements of the blocks by the number of arrangements within the block.

So putting it all together, the total number of arrangements would be 5! * 3! = 120 * 6 = 720. But wait, ......So yes, 5! * 3! = 720. That seems correct.
But let me verify with a smaller example. ...... So the answer should be 720.
Wait, but let me check another way. ...... So total arrangements would be 5! * 5 * 3! = 120 * 5 * 6 = 3600. Wait, that's different. Hmm, so which one is correct?
Wait, this is conflicting. So which approach is right? ......
Wait, now I'm confused. ......
Wait, perhaps the first approach is correct because when we treat the block as a single entity, ......
Wait, but if we fix the 5 people in a certain order, then the gaps are fixed. So inserting the block into a gap would fix the block's position relative to the 5 people......
Predict: 5

**RLVR:** Let's think step by step Okay, so I need to figure out how many ways 8 people can sit around a round table, but with the condition that Pierre, Rosa, and Thomas all want to sit together. And rotations are considered the same, so we don't count those as different. Hmm, let's start by recalling some basics about circular permutations.

First, normally, if you have n people sitting around a round table, the number of ways is (n-1)! because rotations are considered the same. So for 8 people, it would be 7! = 5040. But here, we have a restriction: Pierre, Rosa, and Thomas must all sit together. So how do we handle that?

I remember that when certain people must sit together, we can treat them as a single unit or ̈block.S̈o instead of 8 separate people, we have this block plus the other 5 people. So that's 1 block + 5 people = 6 units total. Then, arranging these 6 units around a circular table. But wait, since it's a circular table, the number of arrangements would be (6-1)! = 5! = 120. But wait, but the block itself can be arranged in different ways, right? Because Pierre, Rosa, and Thomas can be arranged among themselves in 3! ways. So then, total arrangements would be 5! multiplied by 3!.

So that would be 120 * 6 = 720. But wait, hold on. Let me check again. ......
But let me think again. ......
But wait, another way to think about it: ......
Wait, but let me check again. ......
Alternatively, ...... So total is 120*6=720. So yes, that's correct.
Therefore, the answer is 720. So I think that's right.
*Final Answer** 720 </think>
......
Therefore, the final answer is 720 .

