# OpenReview forum: "Self-Healing: Recovering Pruned Large Reasoning Models via Reinforcement Learning"
_ICLR.cc/2026/Conference — Submitted to ICLR 2026_

### Official Review · Reviewer_rY2E · 2025-10-26

**Soundness:** 2
**Presentation:** 3
**Contribution:** 2
**Rating:** 4
**Confidence:** 3

**Summary:**

This paper studies how to recover reasoning performance in structured-pruned large reasoning models (LRMs) and shows that conventional SFT-based recovery can only recover around 80% accuracy and suffers from inefficiency. It introduces reinforcement learning with verifiable reward (RLVR), which enables the pruned LRMs recover over 95%.

**Strengths:**

(1) This paper is clearly written and well-structured, which makes the contribution easy to follow.

(2) The work has potential impact. It demonstrates RLVR is better way to recover reasoning ability of a pruned LLMs.

(3) The empirical evaluation is comprehensive.

**Weaknesses:**

(1)The claim of presenting “the first comprehensive study on the impact of structured pruning on mathematical reasoning ability” seems overstated. Prior work [1] has already investigated the effects of structured pruning on mathematical problem-solving.

(2) The novelty of the paper appears limited. The RLVR method is not new [2], and since it was specifically proposed to enhance LLM reasoning, it is unsurprising that RLVR outperforms SFT in recovering reasoning ability.

(3)The evaluation lacks coverage of strong reasoning models such as GPT-OSS-20B/120B. Most experiments are conducted on models with weak or moderate reasoning ability, which undermines the comprehensiveness of the study. Given that the paper focuses on recovering reasoning ability, it is crucial to ensure that the unpruned baseline models themselves demonstrate strong reasoning performance.

[1] Lele, Nahush, et al. "Rethinking the Value of Training-Free Structured Pruning of LLMs." Transactions on Machine Learning Research.

[2] Shao, Zhihong, et al. "Deepseekmath: Pushing the limits of mathematical reasoning in open language models." arXiv preprint arXiv:2402.03300 (2024).

**Questions:**

(1)How does pruning affect the reasoning ability of strong reasoning LLMs such as GPT-OSS-20B/120B? Furthermore, how effectively does RLVR recover their performance in this setting?

(2) Could the authors provide explanations or insights into why RLVR is more effective than SFT in recovering reasoning ability?

---

### Official Review · Reviewer_nxUJ · 2025-10-27

**Soundness:** 2
**Presentation:** 3
**Contribution:** 2
**Rating:** 4
**Confidence:** 3

**Summary:**

This submission studies training for recovery of performance after pruning a large reasoning model (LRM). The first section of the paper evaluates the performance loss on mathematical reasoning benchmarks due to different pruning methods. The main proposal is to use reinforcement learning with verifiable reward (RLVR) in recovery training, after a first phase of supervised fine-tuning (SFT) the pruned model. Experiments on 5 LRMs and 6 mathematical reasoning benchmarks show that recovery training with RLVR can approach the performance of the unpruned model (which SFT by itself fails to do), while also improving efficiency in terms of reasoning length.

**Strengths:**

- It is good to see evidence that RLVR indeed "breaks the recovery performance bottleneck of SFT," i.e., that RLVR can recover 95-100% of the performance of the unpruned model whereas SFT alone saturates in the 80-90% range.
- Shorter reasoning length (i.e., improved inference efficiency) is a nice additional benefit of RLVR recovery.
- The main paper includes a healthy Discussion section addressing four questions that readers may have.

**Weaknesses:**

1. It is hard for me to get over the fact that the success of reinforcement learning (and RLVR in particular) in training LRMs is well-known, as the submission itself discusses in Section 6.1, lines 441-447. This work basically applies the same idea to recovery training of pruned models.
1. I am not fully convinced of the motivation for this work as embodied by the experimental setup in Section 4.1: pruning 25% of the parameters of 7B-8B parameter models (and one 14B model). Three of the five LRMs are themselves distilled from DeepSeek-R1, which I see as the more significant form of compression. This leads me to think that the following baseline should be added to better put the work in context: First prune an LLM (e.g. LLaMa-3.1-8B-Instruct), and then distill DeepSeek-R1 into the pruned LLM. In the case of Llama-3.1-Nemotron-Nano-8B, which according to its Hugging Face page, underwent a "multi-phase post-training process to enhance both its reasoning and non-reasoning capabilities," the baseline would be to first prune the base LLM and then apply this process. My question is whether first distilling into an 8B model (or applying the Nemotron process to it) and then pruning and recovering is better than directly distilling into a 6B model (or directly applying the Nemotron process). A positive answer to this question would solidly motivate the work.
    - Note that I do not think the experiment in Section 5.2 addresses my question. There the comparison is with Llama-3.1-8B-Instruct, which indeed "lacks prior reasoning ability training." Thus it is more expected that simply applying recovery training to Llama-3.1-8B-Instruct lags behind recovery training of a pruned but well-distilled/well-trained reasoning model. In a similar vein, I do not think the last sentence in Section 5.2 is accurate: "Compared to training a small model from scratch, an LRM obtained via pruning from a larger LRM is more likely to retain stronger reasoning capabilities." The key difference is that the "small model from scratch" is a non-reasoning model versus a well-trained LRM.
1. Section 2.1 claims that structured pruning "does not eliminate the LRM's underlying reasoning capability," and the evidence cited is that the pruned model's pass@$k$ rate approaches the performance of the unpruned model when $k = 64$. However, it is not shown that a model without such reasoning capability, i.e., a non-reasoning model, cannot achieve similar pass@$k$ rates. Thus I do not think the claim is justified (yet).
1. I found the claim that RLVR achieves better data efficiency than SFT to be misleading up until the discussion of Figure 3 in Section 4.2. The reason is that SFT "cold start" (on 94k examples) is part of the proposed recovery training procedure, i.e., the proposed method also incurs the data cost of SFT. What Figure 3 shows is that after this SFT phase, RLVR is more effective and data-efficient than **further** SFT (to 150k or 220k examples). I think therefore that all claims throughout the paper should be revised, to clarify that SFT is part of the proposed procedure and the data efficiency is with respect to further SFT. Furthermore, Figure 1 (right) was especially misleading because it suggests that RLVR is an alternative to SFT, as opposed to a second phase following an SFT phase. Please revise the figure or at least the caption accordingly.

More minor:
1. Several figures (Figure 1 right, Figure 3, Figure 4) are missing labels for the x-axis. Even if the caption somewhat implies what the label should be, it is better to include it. In Figure 1, it is not fully clear what the y-axis label "Average" refers to (average accuracy across the math benchmarks?).
1. It is not clear why the experiment in Section 2.2 uses only LLM-Pruner as the pruning method, since the experiment in Section 2.1 also includes BlockPruner and LLM-Streamline. Similarly in Section 4.1, the choice of LLM-Pruner and LayerDrop is not explained.

Minor:
- Line 136: "a correct answer" --> "the answer is correct" (incomplete sentence)

**Questions:**

In order of importance:
1. My question corresponding to weakness 2 above.
1. In Section 5.4, what do "one-shot pruning" and "two-stage pruning" refer to? I did not see these described previously.
1. To what extent have the "efficient reasoning model" works cited in line 039 explored methods for recovery, for example after quantization?
1. In eq. (1), should $\hat{A_i}$ be indexed only by $i$, or $(i, t)$? If the latter, then I think $r_i$ should also be $r_{i,t}$ in the definition of $\hat{A}_i$?
1. Lines 412-414: Do "50% width pruning and pruning 50% across both width and depth dimensions" correspond respectively to "50% LLM-Pruner" and "50% LLM-Pruner + LayerDrop" in Table 3?

---

### Official Review · Reviewer_wgqE · 2025-10-30

**Soundness:** 3
**Presentation:** 3
**Contribution:** 2
**Rating:** 4
**Confidence:** 3

**Summary:**

The paper proposes to use RLVR method to recover the reasoning performance after structure pruning of reasoning models. This paper shows empirical evidence that RLVR can better recover the mathematical reasoning abilities of LRMs than SFT methods from structured pruning, under specified pruning ratio.

After reading this paper, I am somewhat convinced that using the RLVR method is indeed suitable for the recovery of reasoning models after structured pruning, however the overall significance and effectiveness of the main contributions in paper are unclear, since:

1. Using RL method is nowadays natural in provoking/recovering reasoning capabilities, therefore the methodological novelty is limited. However I do think applying RL to pruning recovery is pioneering.
2. The optimization objective seems only suitable for recovering performance on specific tasks, rather than general reasoning capabilities.

Given the above reasons, my assessment of the paper is weak reject. But I look forward to more insights in the discussion period to help me better understand the contributions of this work.

**Strengths:**

1. The paper focuses on the recoverability of structured pruning of LLMs, and the proposed method potentially have great impact in the field.
2. The results showed in the paper presents strong evidence that the on-policy methods like RLVR outperforms off-policy methods like SFT in recovering pruned models, although it would further strengthen the claims if more experimental results are presented.

**Weaknesses:**

1. Lacking generalization evaluation: The paper only tests RLVR on math benchmarks, and doesn’t not fully investigate the generalization performance of the model after recovery training. This may cause questions on whether the model have catastrophic forgetting after training.
2. The fairness of baseline comparison is unclear: In comparisons between RLVR and SFT, the detailed compute budget is not aligned, since RLVR’s on-policy rollouts add extra computation, which may undermine the claims of efficiency and performance.
3. The paper only presents result on structured pruning results, without discussion on unstructured pruning/recovery (as it may better preserve model capabilities)

**Questions:**

1. The choice of not using a reference model in RLVR procedure: since there are no longer a reference model to regularize the policy, there might be catastrophic forgetting issues, or some degradation to model performance on other tasks. Have the authors evaluated whether this choice leads to degradation in other tasks or general capabilities? How does RLVR compare to standard SFT in terms of generalization and resistance to forgetting?
2. For the baseline comparison, could the authors clarify how compute and data budgets were matched across SFT and RLVR experiments? For example, on-policy rollouts in RLVR would bring more computational overhead, causing unfair comparisons. Outlining these details will improve the clarity of the paper.
3. Have the authors use other RL algorithms (i.e. Dr. GRPO, DAPO) to evaluate their performance in recovering the reasoning capabilities of LRM?
4. The authors use structured pruning for evaluating the method. However, unstructured pruning is often considered to be better in preserving the capabilities of the base model, and intuitively recovering from unstructured pruning is easier. Have the authors tried comparing RLVR and SFT in recovering from models with unstructured pruning?
5. The authors use a SFT-then-RLVR approach to recover the reasoning capabilities after pruning. Could the author provide some insights on why the SFT stage is important in the recovering pipeline, and how it compares with only using RLVR?

---

### Official Review · Reviewer_nooK · 2025-11-01

**Soundness:** 2
**Presentation:** 2
**Contribution:** 2
**Rating:** 2
**Confidence:** 4

**Summary:**

This paper investigates the recovery of mathematical reasoning abilities in LRMs after structured pruning. The core claim is that while structured pruning damages reasoning accuracy, a significant fraction of underlying reasoning capacity persists and can be effectively unlocked through a recovery RLVR.

**Strengths:**

1. This paper presents a good empirical study on how existing pruning methods for LLMs harm reasoning performance in LRMs.
2. Good comparison with SFT, showing that RLVR is a more effective recovery method.

**Weaknesses:**

1. The experimented pruning method, such as LLM-Pruner, are designed for LLM instead of LRMs. Current works on efficient reasoning focus more on pruning the chain-of-thought instead of the model weights. Thus, the authors should validate why using weight pruning methods that are designed for LLMs  in LRMs or find weight pruning methods designed for LRMs as baselines.
2. For the experiment in Figure 2 (a), a fair comparison should also show the pass @ k performance of the original model, instead of showing a straight line.
3. The two models used are all distilled Llama reasoning models. The authors should consider trying other reasoning models, such as the QwQ, Qwen3, or Phi4.
4. All the experiments are done on 8B models; the scalability of this recovery method is unclear.
5. There is no ablation study, which should include a comparison of testing on different percentages of pruning ratios.
6. The authors should mention the computational cost benchmark, such as time consumption and GPU utilization of SFT and the proposed RLVR.
7. All experiments are done on mathematical datasets; an experiment on a dataset in other fields can strengthen the paper by showing the method’s ability to generalize.

**Questions:**

Please refer to the weakness.

---

### Meta-Review · Area_Chair_VuUD · 2025-12-19

**Summary:**

1. Novelty of paper is not enough as using RLVR to enhance LLM reasoning is not new.
2. Experiments are conducted only on 8B-scale models and distilled LLaMA-based reasoning models.
3. Experimental validation is limited to mathematical reasoning datasets.
4. The claim that RLVR achieves better data efficiency than SFT is not convincingly supported by the presented results.
5. The claim of presenting “the first comprehensive study on the impact of structured pruning on mathematical reasoning ability” seems overstated

**Reviewer Concerns:**

No rebuttal was submitted.

**Reviewer Scores:**

As no rebuttal was submitted, the reviewers will not revise their scores.

---

### Decision · Program_Chairs · 2026-01-26

Reject